# Suppressing Uncertainties in Degradation Estimation for Blind Super-Resolution

## ABSTRACT

The problem of blind image super-resolution aims to recover high-resolution (HR) images from low-resolution (LR) images with unknown degradation modes. Most existing methods model the image degradation process using blur kernels. However, this explicit modeling approach struggles to cover the complex and varied degradation processes encountered in the real world, such as high-order combinations of JPEG compression, blur, and noise. Implicit modeling for the degradation process can effectively overcome this issue, but a key challenge of implicit modeling is the lack of accurate ground truth labels for the degradation process to conduct supervised training. To overcome this limitations inherent in implicit modeling, we propose an **U**ncertainty-based degradation representation for blind **S**uper-**R**esolution framework (**USR**). By suppressing the uncertainty of local degradation representations in images, USR facilitated self-supervised learning of degradation representations. The USR consists of two components: Adaptive Uncertainty-Aware Degradation Extraction (AUDE) and a feature extraction network composed of Variable Depth Dynamic Convolution (VDDC) blocks. To extract Uncertainty-based Degradation Representation from LR images, the AUDE utilizes the Self-supervised Uncertainty Contrast module with Uncertainty Suppression Loss to suppress the inherent model uncertainty of the Degradation Extractor. Furthermore, VDDC block integrates degradation information through dynamic convolution. Rhe VDDC also employs an Adaptive Intensity Scaling operation that adaptively adjusts the degradation representation according to the network hierarchy, thereby facilitating the effective integration of degradation information. Quantitative and qualitative experiments affirm the superiority of our approach.

## CCS CONCEPTS

• **Computing methodologies → Image processing**.

## KEYWORDS

Blind Super-Resolution, Learning with Uncertainty, Uncertainty-based Degradation Representation

## 1 INTRODUCTION

Image super-resolution (SR), a highly regarded task within the domain of low-level computer vision, seeks to reconstruct high-resolution (HR) images from their low-resolution (LR) counterparts

*ACM MM, 2024, Melbourne, Australia*
© 2024 Copyright held by the owner/author(s). Publication rights licensed to ACM.
ACM ISBN 978-x-xxxx-xxxx-x/YY/MM
https://doi.org/10.1145/nnnnnnn.nnnnnnn

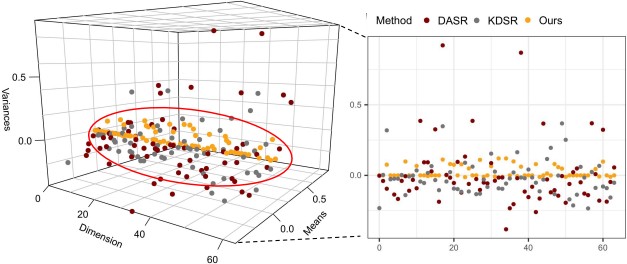

**Figure 1: Comparison on degradation estimation stability. We randomly select different patches from the same image to compare the mean and variance of the degradation representations obtained by DASR, KDSR and USR. These methods exhibit varying degrees of instability, rooted in the inherent uncertainties of the model. USR (ours) demonstrates the most stable performance among them.**

by augmenting the pixel count. This process involves deducing and restoring high-frequency details from a limited array of pixel information, thereby yielding images of enhanced clarity and detail. Conversely, image degradation represents the reverse procedure, wherein LR images are generated from their HR analogs. The degradation process is often unknown and complex, rendering the issue of blind super-resolution a formidable challenge. Modeling the image degradation process aids in reducing the complexity encountered by image SR models.

Traditional super-resolution techniques typically rely on interpolation methods [15, 43]. However, with the advent of deep learning, methods based on neural networks have significantly outpaced traditional approaches. These methods fall into two categories: model-based methods and learning-based methods. Model-based approaches simulate the image degradation process, estimating the degradation mode of LR images before reconstructing the HR images. These methods range from simple to complex, including those based on blur kernel estimation [13], spatially variant blur kernels [27, 66], and implicit modeling [21, 32, 47, 53] of the degradation process. They estimate the degradation mode for each LR image individually, hence are inclined to better generalize across unknown degradations. On the other hand, learning-based methods aim to train a unified super-resolution network using a vast corpus of LR/HR image pairs synthesized based on presumed degradation models [37, 64, 68]. Yet, these learning-based approaches are heavily dependent on the training data and may suffer significant performance drops when there is a domain discrepancy between the training and testing data [49]. Some efforts attempt to simulate real-world degradation patterns with more complex training samples, notable among which are BSRGAN [63] and Real-ESRGAN [48], employing advanced degradation models that incorporate blur, noise, resizing, and JPEG compression to generate training samples.

Researchers widely regard the challenge of addressing such complex degradation processes as blind super-resolution [42, 52, 53].

Model-based approaches predominantly rely on blur kernels. However, these methods possess limited representational capacity and can only cover blur-related degradation, falling short in the face of noise, JPEG compression, and other complex degradation processes. To address the nearly infinite degradation modes in blind super-resolution tasks, recent works have proposed using implicit modeling to characterize degradation patterns. DASR [47] employs contrastive learning to distance or draw closer feature representations of different degradation modes, whereas KDSR [53] uses knowledge distillation to enable a student network to learn the degradation representation from a teacher network.

As illustrated in Figure 1, DASR and KDSR exhibit instability in estimating degradation representations, meaning they fail to obtain consistent degradation representations for the same LR image. Such instability and inaccuracies in degradation representation adversely affect subsequent super-resolution processes. This instability is a manifestation of model uncertainty [11]. The root causes of the instability and unreliability in the degradation features of these methods are: (1) Due to the absence of ground truth, these methods provide only coarse constraints on the estimation process. (2) They overlook the uncertainty present in estimating implicit degradation representations, failing to offer confidence or uncertainty estimates for the generated outcomes.

To address this issue, we introduce an Uncertainty-based degradation representation for blind Super-Resolution (USR) framework. To quantify and mitigate the uncertainty in Uncertainty-based Degradation Representation (UDR) estimation, we constrain UDR with Self-supervised Uncertainty Contrast which suppress the uncertainty of local degradation representations in images. Furthermore, to ensure effective guidance of UDR, we have designed Variable Depth Dynamic Convolution (VDDC) Block. Thorough experimentation validates the efficacy of our proposed modules. In summary, our contributions are as follows:

- We introduce the framework named **U**ncertainty-based degradation representation for blind **S**uper-**R**esolution (**USR**). USR initially obtains Uncertainty-based Degradation Representation (UDR) from LR images through implicit modeling. To fully leverage the UDR, we propose the Variable Depth Dynamic Convolution (VDDC) Block. With dynamic convolution and Adaptive Intensity Scaling (AIS) of UDR, VDDC effectively integrates image degradation information.

- We introduce the Adaptive Uncertainty-Aware Degradation Extraction (AUDE). Within AUDE, our proposed Self-supervised Uncertainty Contrast module employs USLoss to self-supervisedly constrain and mitigate the uncertainty inherent in the UDR estimation process. This approach not only addresses the challenge of implicit representations lacking true values but also enhances the model's ability to adeptly handle various degradation modes. To our knowledge, we are the first to propose uncertainty modeling of the implicit representation for the image degradation process.

- Extensive experiments conducted on multiple representative datasets have demonstrated the performance of USR. A comprehensive suite of qualitative experiments, quantitative analyses,

and ablation studies underscores the efficacy of our proposed modules.

## 2 RELATED WORK

### 2.1 Blind Super-Resolution

Contrary to the traditional Single Image Super-Resolution (SISR) task, the objective of blind super-resolution is to reconstruct HR images from their LR equivalents without prior knowledge of the degradation process [33, 44, 60]. Blind super-resolution methods can generally be categorized into the following two types.

**Model-based SR.** This category of SR models the image degradation process. Most methods employ explicit modeling based on Equation (1), where $k$ represents the blur kernel, and $s$ denotes the downsampling factor. Within the realm of methods based on blur kernel estimation, IKC [13] introduced an iterative estimation technique and designed a correction function to accurately estimate the blur kernel or degradation features. Utilizing the principle of internal cross-scale recurrence, KernelGAN [5] interprets the maximization of patches within a single image as a problem of data distribution learning and trains a Generative Adversarial Network (GAN) across patches. MANet [27], a method based on the estimation of spatially variant blur kernels, estimates blur kernels with a network designed to have an optimally sized receptive field. However, these approaches struggle to address degradation modes beyond blur. In the domain of methods based on implicit degradation modeling, DASR[47] and KDSR [53] characterize the image degradation process using contrastive learning and knowledge distillation, respectively. However, due to the lack of effective constraints, DASR and KDSR are unstable and cannot fully extract discriminative degradation representations to guide blind super-resolution.

$$LR = (HR \otimes k) \downarrow_s + n \qquad (1)$$

**Learning-based SR.** Learning-based methods endeavor to directly learn degradation patterns from training data in the form of high-level semantics, foregoing modeling the degradation process [28]. SwinIR [26], by adopting the Swin Transformer for image restoration tasks, has achieved breakthrough performance. Works such as Restormer [61], HAT [8], and DAT [9] further demonstrate the potential of Vision Transformers in low-level visual tasks. Additionally, some researchers have focused on diffusion models [38, 45, 50, 59], which transform complex and unstable generative processes into several independent and stable reverse processes through Markov chain modeling. However, due to the inherent randomness of probabilistic models, images generated from the sampling space by diffusion models diverge from real images. Nonetheless, these efforts typically excel only within data distributions identical to their training sets, displaying limited generalizability.

### 2.2 Modeling Uncertainty for Super-Resolution

**Uncertainty in Deep Learning.** As deep learning continues to evolve, neural networks have permeated nearly every scientific domain, becoming integral to a myriad of real-world applications [10, 17, 20, 29, 57]. Researchers have devoted significant effort to understanding and quantifying the uncertainty in neural network predictions to enhance the performance and robustness of deep

networks [35, 55, 58, 62]. In the field of computer vision, uncertainty modeling plays a pivotal role in critical tasks such as image classification [40, 46], object detection [16], semantic segmentation [4], face recognition [65, 67], action recognition [41, 62], and image generation [39]. The uncertainty in deep learning can broadly be categorized into two types: data uncertainty (also known as aleatoric uncertainty), which describes the intrinsic noise within the data, and model uncertainty (also known as epistemic uncertainty), which reflects the uncertainty inherent in the model itself due to inadequate training, insufficient training data, and other factors.

As shown on Equation (2) , given a parameterized model $f(\theta)$, model uncertainty is formalized as a probability distribution over the model parameters $\theta$, while data uncertainty is formatted as a probability distribution over the model output $y^*$. The term $p(\theta|D)$ is referred to as the posterior distribution of model parameters. $D$ indicates the training dataset [11]. Our work focuses on model uncertainty within image degradation representation.

$$p(y^*|x^*, D) = \int \underbrace{p(y^*|x^*, \theta)}_{\text{Data}} \underbrace{p(\theta|D)}_{\text{Model}} \, d\theta \qquad (2)$$

**Unvertainty-based SR.** To date, only a limited number of studies have explored the potential of uncertainty modeling in the task of super-resolution. SOSR [2] investigated the issue of source-free domain adaptation within super-resolution tasks through uncertainty modeling. DDL [30] utilized Bayesian methods to assess the reliability of high-frequency inference from a frequency domain perspective. [36] introduced an uncertainty-driven loss function that incorporates per-pixel uncertainty into super-resolution, giving priority to pixels with greater certainty, such as those representing texture and edges. Another study employed batch normalization uncertainty to analyze super-resolution uncertainty, thereby enhancing network robustness against adversarial attacks [22]. GRAM [25] focused neural network attention on challenging images through Gradient Rescaling Attention. However, none of these efforts addressed the blind super-resolution challenge associated with complex degradation processes.

## 3 METHOD

### 3.1 Overview

As previously mentioned, implicit modeling of complex and variable degradation processes represents a promising approach to addressing the issue of image blind super-resolution, with the lack of ground truth posing a significant challenge. To tackle this challenge, we employs an Uncertainty-based Degradation Representation (UDR) to model various degradation processes, adapting to complex and varied degradation scenarios.

As illustrated in Figure 2 (b), we implement Adaptive Uncertainty-Aware Degradation Extraction (AUDE) on LR images. In detail, we obtain the UDR from LR images using the Degradation Extractor (DE). To facilitate self-supervised training of UDR, we have devised the USLoss and an Self-supervised Uncertainty Contrast module. As depicted in Figure 2 (c), to effectively harness the information encapsulated within the UDR, we have designed a Variable Depth Dynamic Convolution (VDDC) Block, which facilitates the modulation of UDR intensity in accordance with the depth of the network.

Overall, we utilize the DE to derive the UDR from LR images. Concurrently, the LR image undergoes initial shallow feature extraction via a convolutional layer, followed by deep feature extraction through $N$ VDDCs.

### 3.2 Adaptive Uncertainty-Aware Degradation Extraction

Within AUDE, DE is tasked with extracting UDR from LR images. To suppress the uncertainty in the DE network, we employed USLoss within the Self-supervised Uncertainty Contrast to conduct self-supervised training of DE.

**Degradation Extractor.** Acknowledging the limitations of blur kernel modeling, which is confined to addressing solely blur-related degradation processes, we have adopted a more potent approach of implicit modeling for a comprehensive characterization of the degradation process. As depicted in Figure 2 (b), the DE extract degradation representations from LR images adaptively.

Initially, DE subjects the image to a preliminary feature extraction phase involving a $3 \times 3$ convolutional layer followed by a ReLU layer. This is succeeded by the application of multiple convolutional blocks tasked with the extraction of deeper features. Each convolutional block is systematically composed of a $3 \times 3$ convolutional layer, a ReLU layer, and another $3 \times 3$ convolutional layer, arranged in sequence. The final stage of this process involves the refinement of the degradation representation vector through an average pooling layer and a Multilayer Perceptron (MLP).

Within the process of AUDE, during the training phase, two distinct patches are obtained from a LR image. The DE is employed to extract the corresponding UDR from these patches. By applying USLoss to suppress the uncertainty associated with these two UDRs, we guide the DE towards more stable degradation estimations. During inference, DE directly extracts a global UDR from the entire LR image.

**Self-supervised Uncertainty Contrast.** While different parts of the same image undergo nearly identical degradation, as illustrated in Figure 2 (b), the DE estimates significantly varied degradation representations from different patches of a LR image. This discrepancy reveals that without meticulous constraints, the UDR derived by DE is inconsistent and unstable, thus incapable of furnishing precise degradation information for subsequent SR networks. Ideally, however, the UDR obtained from different patches or the entire image should exhibit consistency. Our objective is for DE to estimate a unified and accurate degradation representation, both locally and globally. To address this challenge, we have designed a USLoss and an Self-supervised Uncertainty Contrast module to mitigate this uncertainty, enabling DE to estimate degradation representations more stably and accurately.

Let $u_1$ and $u_2$ represent the degradation representations obtained from two different patches $x_1$ and $x_2$ within a LR image. From probabilistic perspective, our training objective aims to maximize the joint distribution probability in Equation (3):

$$P(u_1|x_1; W, u_2|x_2; W) = P(u_1|x_1; W) \cdot P(u_2|x_2; W) \qquad (3)$$

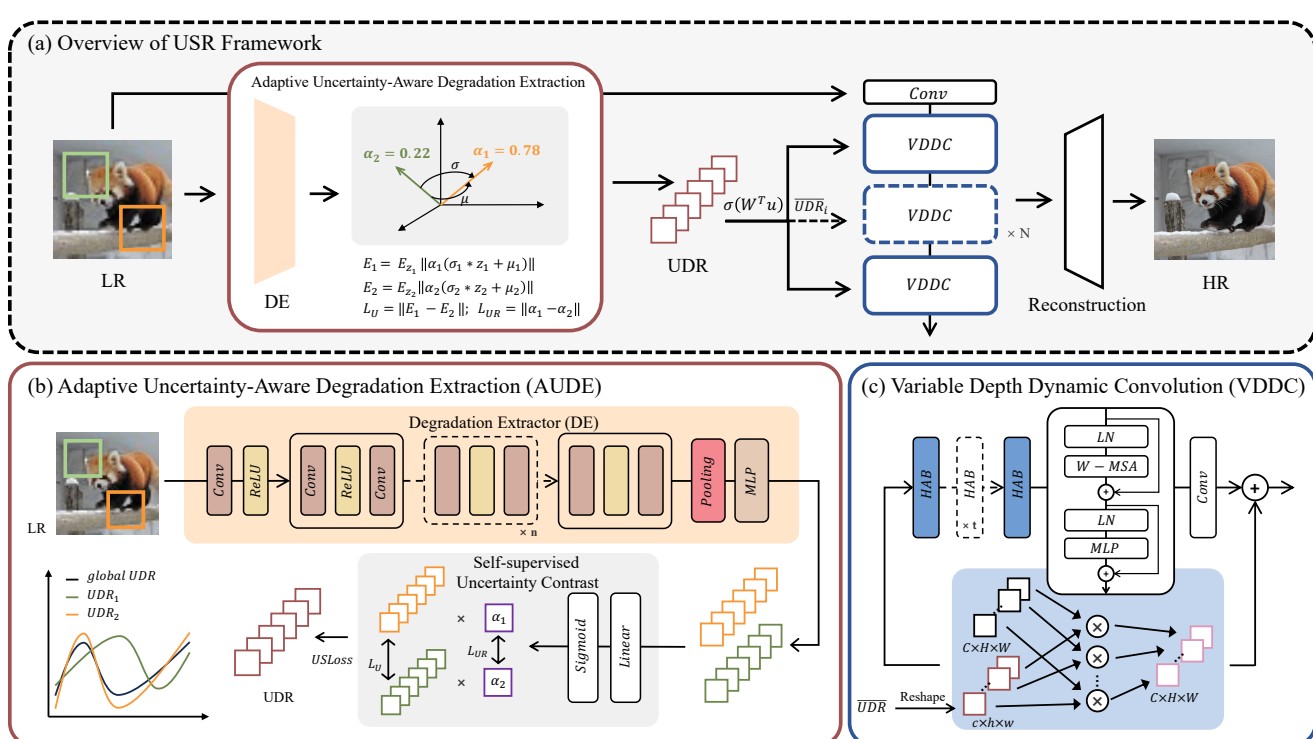

**Figure 2: The proposed framework Uncertainty-based degradation representation for blind Super-Resolution (USR). (a) Illustration of the main process of USR. USR extracts Uncertainty-based Degradation Representation (UDR) from the LR image, which is then integrated with the super-resolution process through the VDDC. Reconstruction refers to the process of upsampling features. (b) Depiction of the Adaptive Uncertainty-Aware Degradation Extraction (AUDE). AUDE trains the Degradation Extractor (DE) in a self-supervised manner. (c) Depiction of the Variable Depth Dynamic Convolution (VDDC) Block. VDDC integrates UDR while extracting deep features from the LR image.**

where $W$ represents the parameters of the model. To maximize the aforementioned objective, we should aim to minimize the negative log likelihood of the joint distribution probability, where the random variables are $u_1$ and $u_2$. Therefore, the optimization objective transforms into Equation (4):

$$E_{u_1,u_2 \sim P(u_1|x_1;W),P(u_2|x_2;W)} \left[ P(u_1, u_2) \right] \quad (4)$$

To address this probability distribution, we model $P(u|x;W)$ as a multivariate normal distribution, drawing upon the Central Limit Theorem (CLT) [24] and non-local means [7]. When there are sufficiently many random variables, their sum or average tends toward a normal distribution. This leads to Equation (5):

$$P(u|x;W) \sim N\left(\mu_{(x;W)}; \Sigma_{(x;W)}\right) \quad (5)$$

The mean $N$ and covariance $\Sigma$, outputs of the network parameterized by $W$, form the crux of our approach. However, the expectation in Equation (4) necessitates sampling $u$ from $P(u_1, u_2)$, an operation intrinsically non-differentiable. To facilitate backpropagation through $u$, we employ reparameterization [23] to transfer the sampling process to the stochastic variable $z \sim N(0, 1)$, culminating in Equation (6):

$$E_{z_1,z_2 \sim N(0,1)} \left[ P(\mu_1 + \sigma_1 z_1 | \mu_2 + \sigma_2 z_2) \right] \quad (6)$$

In accordance with the conditional model delineated by the Gibbs distribution [6, 12], we arrive at Equation (7):

$$P(u_1, u_2) \propto \prod_{i}^{h \times w} \exp\left(-\frac{|u_1 - u_2|}{kT}\right) \quad (7)$$

The corresponding Gibbs energy is expressed in the form $|u_1 - u_2|$, with $kT$ denoting the constant partition function [14]. Hence, we derive Equation (8):

$$\min_{E_{z_1,z_2}} \left[ \frac{1}{kT} \sum_{i}^{h \times w} |(\mu_{i1} - \mu_{i2}) + (\sigma_{i1} z_1 - \sigma_{i2} z_2)| \right] \quad (8)$$

Equation (8) impartially treats patches from different regions of an image. However, due to the inherent model uncertainty present in the feature extraction process of DE, the degradation representations extracted from different patches are not stable [10, 11, 48]. Aiming to direct DE towards a representation of degradation with diminished uncertainty, we have conceptualized the Self-supervised Uncertainty Contrast module. Within this module, a sequence of a linear layer and a Sigmoid layer is employed to deduce a learnable

variable, the Uncertainty Aware weight $\alpha$. This weight is then multiplied by the degradation representation to adaptively modulate its expressive intensity. Furthermore, we refine Equation (8), orienting it towards a suppression of the uncertainty estimated by DE in the direction of lower uncertainty, yielding Equation (9):

$$L_U = E_{z_1,z_2}\left[\frac{1}{kT}\sum_i^{h\times w}|\alpha_1\left(\mu_{i1}+\sigma_{i1}z_1\right)-\alpha_2\left(\mu_{i2}+\sigma_{i2}z_2\right)|\right] \quad (9)$$

To prevent the DE from degrading to a local optimum during the training process, we incorporate a regularization term based on the Uncertainty Aware weight $\alpha$. This compels the Self-supervised Uncertainty Contrast module to discriminate the uncertainty among different patches, as indicated in Equation (10):

$$L_{UR} = |\alpha_1 - \alpha_2| \quad (10)$$

In summary, the final USLoss is encapsulated in Equation (11), where $\lambda$ represents scaling weight.

$$USLoss = L_U - \lambda L_{UR} \quad (11)$$

Through Equation (11), we have mitigated the uncertainty inherent in the DE estimation process of degradation representation, which we term as Uncertainty-based Degradation Representation (UDR).

## 3.3 Variable Depth Dynamic Convolution Block

Through uncertainty modeling, we acquire the UDR via the Degradation Extractor (DE). To leverage UDR to its fullest extent, we have crafted the Variable Depth Dynamic Convolution (VDDC) Block, which adjusts the intensity of UDR based on the depth of the network and efficiently mines the feature information of images.

In the feature extraction segment, we adopt the design from HAB [8], wherein the channel attention within HAB and the design of the window-based multi-head self-attention [31] have been proven to effectively extract features. Each VDDC, in addition to $t$ HABs, also includes a residual group composed of a layer normalization, a W-MSA layer [31], another layer normalization and an MLP, as well as a $3 \times 3$ convolutional layer for fine-tuning the features.

As illustrated in Figure 2 (c) and inspired by KDSR [53] and UDVD [54], we integrate UDR using dynamic convolution. We first reshape UDR to the dimensions of $c \times h \times w$, and let $F$ denote a feature map with dimensions $C \times H \times W$. For each channel $i$, the convolution output $O_i$ at position $(x, y)$ can be described by Equation (12).

$$O_i(x,y) = \sum_{m=0}^{h}\sum_{n=0}^{w}F_i(x+m,y+n)\cdot u_i(m,n) \quad (12)$$

where $u$ represents the dynamic convolution weights, and $O$ denotes the output features with dimensions $C \times H \times W$. However, in such a feature fusion approach, the UDR introduced at different depths of the network remains constant. A more rational approach would involve adaptively adjusting the intensity of the UDR input based on the network hierarchy. Thus, inspired by the concept of [46], we perform an Adaptive Intensity Scaling operation (AIS) on UDR before the dynamic convolution, as illustrated in Equation (13).

$$\overline{UDR}_i = \gamma_i \times UDR \quad (13)$$

The adaptive scaling parameter $\gamma$ is derived as Equation (14).

$$\gamma_i = \sigma_i\left(W^T u + b\right) \quad (14)$$

where $\sigma$ refers to activation function, $W$ represents the transformation matrix and $b$ represents the linear bias.

## 4 EXPERIMENT

### 4.1 Experiment Setup

**Implementation details.** In the DE, the number of convolutional blocks is set to 5; the MLP consists of three linear layers and three LeakyReLU layers in alternation. The scaling weight $\lambda$ in USLoss is set to 0.1. USR incorporates 7 VDDC blocks, each containing 6 HABs. For the activation function in Equation (14), we have chosen the Sigmoid. The MLP within the VDDC adheres to the configuration specified in [31]. The final Reconstruction segment comprises a convolutional layer, a pixel shuffle layer, and another convolutional layer to upsample the feature map into an image.

USR is trained on a mixed dataset comprising DIV2K and Flickr2K. The training process is divided into three stages: (1) We train the VDDC to extract image features using MSE Loss; (2) Subsequently, by leveraging USLoss to suppress model uncertainty, we undertake self-supervised training of the DE; (3) Finally, the network is fine-tuned using L1 Loss. More Implementation details will be offered in the **supplementary materials**.

**Testing datasets.** We conducted comparisons between USR and several representative methods across six widely used datasets: DIV2K [1], BSDS [3], Urban100 [18], T91 [56], DPED [19] and DRealSR [51]. Synthetic data were generated following the workflow proposed by Real-ESRGAN [48].

### 4.2 Comparison With Existing Methods

**Compared Methods.** To evaluate the effectiveness and performance of our method, we compared USR with current state-of-the-art and representative blind super-resolution approaches, including DAN [34], DCLS [33], DASR [47], MANet [27], KDSR [53], SwinIR [26], HAT [8], Real-ESRGAN [48], and ResShift [59]. We tested these methods using their officially available codes.

**Quantitative Comparisons.** Quantitative results from the Table 1 reveal USR's significant advantages across multiple datasets, particularly in the image super-resolution domain. At a $\times 4$ magnification factor, USR achieved the highest PSNR and SSIM on nearly all datasets.

On the DIV2K validation set, USR reached a PSNR of 23.96 dB, surpassing other methods, and led in SSIM with a score of 0.78, demonstrating its exceptional capability in restoring high-quality images. Similarly, on the BSDS100 and Urban100 datasets, USR not only led in PSNR with scores of 29.89 dB and 22.53 dB, respectively, but also achieved the highest SSIM scores of 0.92 and 0.77, further proving its robust performance across different types of images.

Notably, on the DRealSR dataset, even when facing complex real-world scenes, USR maintained a high level of performance with a PSNR of 31.02 dB, slightly higher than other methods. This result is particularly significant considering the test's closer alignment with

**Table 1: Quantitive results on DIV2K, BSDS100, Urban100, T91, DPED and DRealSR datasets for scaling factor ×4. Bold indicates the best performance.**

| Method / Datasets | | DAN | DCLS | DASR | MANet | KDSR | SwinIR | HAT | RealESRGAN | ResShift | USR (Ours) |
|---|---|---|---|---|---|---|---|---|---|---|---|
| DIV2K [1] | PSNR | 22.17 | 22.41 | 21.45 | 18.95 | 22.79 | 22.08 | 22.01 | 22.23 | 22.38 | **23.96** |
| | SSIM | 0.72 | 0.74 | 0.67 | 0.60 | 0.75 | 0.73 | 0.72 | 0.73 | 0.72 | **0.78** |
| BSDS100 [3] | PSNR | 27.95 | 28.03 | 28.79 | 22.79 | 27.50 | 26.08 | 28.04 | 26.62 | 26.40 | **29.89** |
| | SSIM | 0.87 | 0.89 | 0.89 | 0.77 | 0.88 | 0.84 | 0.87 | 0.85 | 0.81 | **0.92** |
| Urban100 [18] | PSNR | 20.26 | 21.18 | 21.36 | 17.05 | 21.28 | 20.37 | 19.56 | 20.61 | 21.71 | **22.53** |
| | SSIM | 0.69 | 0.72 | 0.73 | 0.54 | 0.74 | 0.71 | 0.67 | 0.72 | 0.74 | **0.77** |
| T91 [56] | PSNR | 33.14 | **33.82** | 33.64 | 27.24 | 30.30 | 29.41 | 33.49 | 29.82 | 27.93 | 31.21 |
| | SSIM | 0.93 | 0.93 | 0.94 | 0.90 | 0.94 | 0.92 | 0.93 | 0.93 | 0.85 | **0.95** |
| DPED-blackberry [19] | PSNR | 22.96 | 23.38 | 23.54 | 20.22 | 22.92 | 22.04 | 22.75 | 21.89 | 22.49 | **24.20** |
| | SSIM | 0.74 | 0.76 | 0.76 | 0.64 | 0.75 | 0.72 | 0.74 | 0.72 | 0.71 | **0.78** |
| DPED-iphone [19] | PSNR | 25.52 | 26.05 | 26.00 | 21.08 | 24.88 | 23.79 | 25.43 | 23.71 | 24.37 | **26.77** |
| | SSIM | 0.82 | 0.84 | 0.84 | 0.71 | 0.82 | 0.80 | 0.82 | 0.79 | 0.79 | **0.86** |
| DPED-sony [19] | PSNR | 20.43 | 20.90 | 21.02 | 18.95 | 20.98 | 20.72 | 20.20 | 20.56 | 20.86 | **23.92** |
| | SSIM | 0.64 | 0.66 | 0.66 | 0.57 | 0.66 | 0.65 | 0.64 | 0.65 | 0.63 | **0.69** |
| DRealSR [51] | PSNR | 31.00 | 30.97 | 30.98 | 27.42 | 29.89 | 28.45 | 30.96 | 29.94 | 25.77 | **31.02** |
| | SSIM | 0.92 | 0.92 | 0.92 | 0.90 | 0.91 | 0.89 | **0.93** | 0.92 | 0.71 | 0.91 |

real-world applications. Although slightly below the highest SSIM score, USR's performance remains impressive given the complexity of real scenarios.

Overall, USR's consistently high performance across various datasets highlights its remarkable advantages in the field of image super-resolution, especially in handling real-world images. Comparing different methods shows that USR excels not only in traditional evaluation standards but also in adapting to and managing complex real-world scenes.

**Qualitative Comparisons.** As illustrated in Figure 3, due to the inherent challenges of the blind super-resolution task, super-resolution models encounter issues such as distortion and artifacts. Compared to other methods, USR excels in preserving the authenticity of the original image and restoring details. Observing the HR image alongside the image processed by USR, one can clearly see USR's significant advantage in maintaining the overall structure and color fidelity of the image.

In the case of stained glass, USR not only faithfully preserves the color and sheen of the glass but also demonstrates superior clarity in edges and details compared to other methods. Particularly in the intricate depiction of the stained glass's central pattern, USR reveals detail levels and color gradients close to the original, whereas other methods exhibit some distortion in these aspects.

In urban street scene case, USR similarly showcases its strengths. Observing the window frames and the brickwork on walls, USR's precision in detail restoration is evident. In contrast, other methods are either too blurry, losing some details, or too sharp, resulting in unnatural artifacts along edges. USR achieves a good balance in handling these details, restoring true textures and depth, making the image closer to the original high-resolution version.

In summary, USR not only provides a more natural and smooth overall visual experience but also maintains a high fidelity in restoring everything from subtle textures to macro structures. Its performance surpasses other super-resolution methods in several aspects, whether it's in detail sharpening, color accuracy, or the naturalness in avoiding over-processing. More Qualitative comparision results will be offered in the **supplementary materials**.

### 4.3 Ablation Study

**Effectiveness of AUDE and AIS.** As shown in Table 2, we conducted experiments across multiple datasets to validate the effectiveness of AUDE and AIS. AUDE is a crucial component of USR, achieving the implicit representation of the image degradation process; AIS is a vital element of VDDC, performing adaptive intensity adjustments to UDR based on the network hierarchy.

Evaluation results on three distinct datasets—DIV2K, BSDS100, and Urban100—demonstrate that the simultaneous application of AUDE and AIS yields the highest PSNR and SSIM scores. Notably, on the DIV2K dataset, PSNR and SSIM reached 23.96 and 0.78, respectively; on the BSDS100 dataset, scores were 29.89 and 0.92, respectively; and on the Urban100 dataset, the scores were 22.53 and 0.77. This starkly contrasts with results obtained using AUDE or AIS alone, where the performance with just AUDE outperforms that with only AIS.

As shown in Table 3, we also analyzed the performance with different numbers of VDDC blocks $N$. The findings reveal that an optimal performance is achieved with 7 VDDC blocks (our approach), manifesting in a PSNR of 23.96 and a SSIM of 0.78 on DIV2K; a PSNR of 29.89 and a SSIM of 0.91 on BSDS100; and a PSNR of 22.53

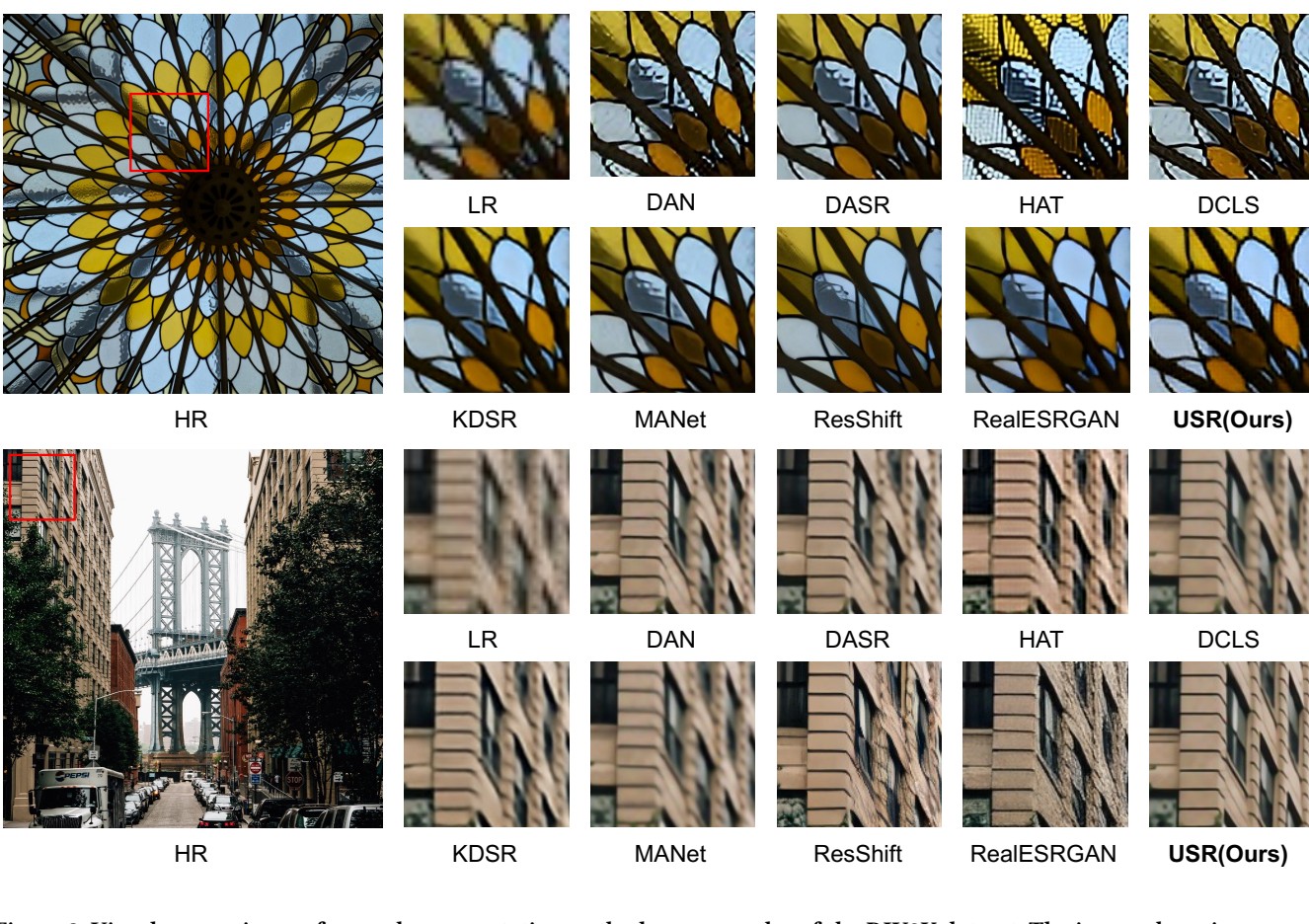

**Figure 3: Visual comparisons of several representative methods on examples of the DIV2K dataset. The image above is a case of stained glass, while the image below depicts a urban street scene.**

**Table 2: Ablation study on the proposed Adaptive Uncertainty-Aware Degradation Extraction (AUDE) and Adaptive Intensity Scaling (AIS).**

| AUDE | AIS | DIV2K | | BSDS100 | | Urban100 | |
|---|---|---|---|---|---|---|---|
| | | PSNR | SSIM | PSNR | SSIM | PSNR | SSIM |
| ✓ | ✗ | 22.85 | 0.67 | 28.71 | 0.82 | 21.55 | 0.67 |
| ✗ | ✓ | 16.88 | 0.52 | 19.35 | 0.64 | 16.69 | 0.51 |
| ✓ | ✓ | **23.96** | **0.78** | **29.89** | **0.92** | **22.53** | **0.77** |

**Table 3: Performance on DIV2K, BSDS100 and Urban100 datasets for different VDDC number $N$.**

| Number of VDDC | DIV2K | | BSDS100 | | Urban100 | |
|---|---|---|---|---|---|
| | PSNR | SSIM | PSNR | SSIM | PSNR | SSIM |
| 6 | 23.20 | 0.76 | 28.66 | 0.90 | 21.66 | 0.73 |
| 7 (Ours) | **23.96** | **0.78** | **29.89** | **0.91** | **22.53** | **0.77** |
| 8 | 23.22 | 0.76 | 28.68 | 0.89 | 21.71 | 0.73 |
| 10 | 23.65 | 0.77 | 29.31 | 0.90 | 22.18 | 0.76 |

and SSIM of 0.77 on Urban100. In contrast, configurations featuring alternative quantities of VDDC blocks experience a marginal decline in performance.

**Effectiveness of USLoss.** Figure 4 presents ablation experiments on USLoss. From left to right, the images display the LR image, HR image, and the results processed by USR under three different configurations. In Figure 4 (a), the reconstructed image exhibits noticeable color distortions in certain areas, particularly in the diagonal sections of the image, where purple and blue spots and stripes are visible, significantly differing from the original high-resolution image's tones. This distortion likely results from the

loss function's lack of constraints on model estimation uncertainty. In Figure 4 (b), beyond color distortion, there are also structural distortions and geometric distortions in the image. Observing the seams of diagonals and wall corners, it is apparent that lines have been inaccurately reconstructed, leading to bending and twisting, contrasting with the original image's straight lines and sharp edges. These structural distortions indicate that without $L_{UR}$, USR falls short in maintaining image geometric integrity and edge clarity. Lastly, Figure 4 (c) showcases the USR method employing both $L_U$ and $L_{UR}$ (our method), where, in this scenario, PSNR significantly increases to 22.11, and SSIM to 0.86. This demonstrates that the

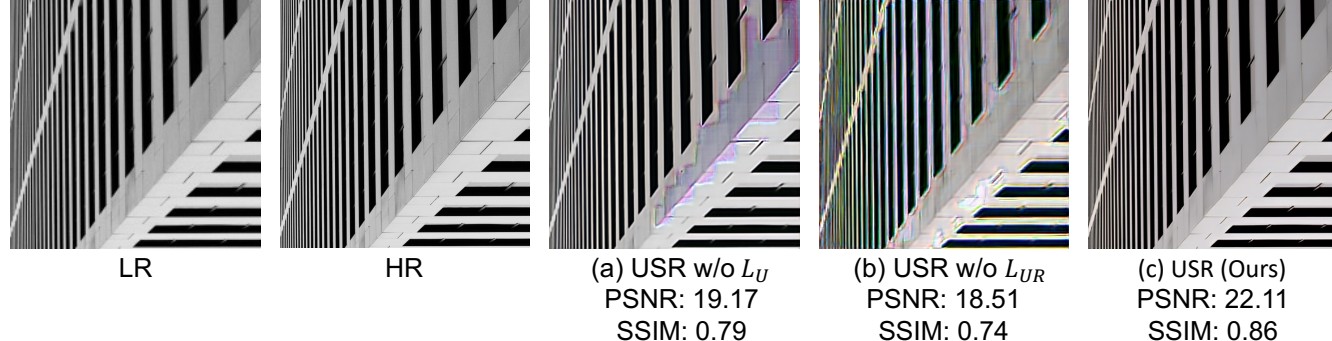

Figure 4: Visualization of ablation study on USLoss. $L_U$ and $L_{UR}$ represent the two components of USLoss. (a) represents USR trained without $L_U$; (b) depicts USR trained without $L_{UR}$; (c) shows USR trained with USLoss.

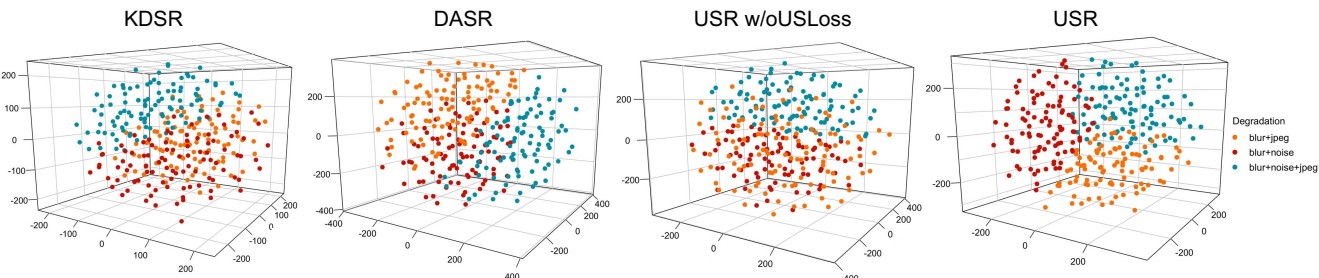

Figure 5: The t-SNE visualizations on the DIV2K datasets. Blur, noise, and JPEG compression represent common degradation modes in real-world scenarios. We conducted cluster analysis experiments under their various combinations. USR (Ours) effectively distinguishes between different degradation modes.

USR method, incorporating both types of loss functions, can better restore image details, closely matching the HR image.

**t-SNE visualization of degradation representation.** Figure 5 uses the t-SNE visualization to demonstrate how four SR algorithms manage various image degradations, with each dot color denoting a different degradation cluster. USR's plot shows tightly clustered points for each degradation mode, highlighting its effective differentiation, while USR without USLoss shows dispersed clusters, underscoring USLoss's importance. KDSR exhibits moderate clustering, less distinct than USR, and DASR shows the most scattered distribution, indicating its lower effectiveness with mixed degradations. These results underscore USLoss's crucial role in improving SR algorithms' ability to discern between degradation modes.

**Comparision on various degradation modes.** As shown in Table 4, experiments conducting quantitative comparisons across various degradation modes on the DIV2K dataset demonstrate the exceptional generalization capability of the USR algorithm. USR achieved the highest PSNR and SSIM values across all considered combinations of degradation modes (blur+noise+JPEG, blur+noise, blur+JPEG). Notably, in the composite degradation scenario of blur, noise, and JPEG compression, USR led with a PSNR of 23.96 and an SSIM of 0.78, significantly outperforming other methods. This underscores USR's outstanding adaptability and robustness in handling multiple degradation effects, effectively enhancing image

quality and maintaining high performance even amidst complex interwoven degradation modes.

Table 4: Quantitative comparison of different methods under various degradation modes on the DIV2K dataset.

| Method | blur+noise+JPEG | | blur+noise | | blur+JPEG | |
|---|---|---|---|---|---|---|
| | PSNR | SSIM | PSNR | SSIM | PSNR | SSIM |
| DASR | 21.45 | 0.67 | 23.04 | 0.75 | 23.07 | 0.75 |
| KDSR | 22.79 | 0.75 | 22.79 | 0.75 | 22.78 | 0.75 |
| USR | **23.96** | **0.78** | **23.19** | **0.76** | **23.24** | **0.76** |

## 5 CONCLUSION

In summary, our proposed Uncertainty-based Super-Resolution (USR) framework effectively addresses the challenge of blind image super-resolution by leveraging implicit modeling. Through Adaptive Uncertainty-Aware Degradation Extraction (AUDE) and the Self-supervised Uncertainty Contrast module, USR accurately extracts degradation information and facilitates self-supervised training. In future work, we aim to further investigate how to address the complex and varied degradation processes in image super-resolution tasks through more refined modeling approaches. Additionally, we aspire to delve deeper into the potential of uncertainty learning within low-level vision tasks.

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
