# OpenReview forum: "Suppressing Uncertainties in Degradation Estimation for Blind Super-Resolution"
_acmmm.org/ACMMM/2024/Conference — MM2024 Poster_

### Official Review · Reviewer_XW5i · 2024-05-23

**Rating:** 4
**Confidence:** 4

**Summary:**

For the Blind Image Super-Resolution task, this work proposes the Uncertainty-based Super-Resolution (USR) framework by using implicit modeling, which includes two core modules: Adaptive Uncertainty-Aware Degradation Extraction (AUDE) and Self-Supervised Uncertainty Contrast module. The former aims at extracting degradation representations, while the latter aims at adding constraints to degradation representations via uncertainty learning. Finally, the accurate degradation representation is used to guide the network reconstruction.

**Strengths:**

The paper is well-organized, ensuring it is easily understandable.

**Limitations:**

(1) Figure 2 is not easy to understand. My understanding is as follows:

- For (a), the input LR extracts initial features through Conv, and then the initial features and UDR are fed into the cascaded VDDC for reconstruction. Then the line from LR to Conv should not be placed under AUDE, which is easy to confuse. Finally, is the UDR fed into the VDDC during training based on the global image?
- For (b), the expression in the figure means that a and u are multiplied point-wise and then the UDR is obtained by USLoss. Is there any - ambiguity with the expression in the article? My understanding from reading the article is that DE gets the degradation representation UDR after the MLP layer. To mitigate the uncertainty inherent in the degradation representation in the DE estimation process, two image patches are randomly selected for self-supervised uncertainty comparison learning. Finally, the UDR is fed to the VDDC after adaptive adjustment via Equation 13. So, in equation 13, u represents the dynamic convolutional weights, i.e. the degradation representation UDR?
- For (c), where are the features of the input HAB? Is the result of the dot product of the image features and UDR added at the end of the module? Also, u already represents the degradation mapping of the image patch (L341), it is better not to use u to represent the dynamic convolutional weights (L514). If both represent the same thing, it should be specified in L514.

(2) I have a small question about the motivation for this paper:
- The following is from L327-339 in the paper：
“While different parts of the same image undergo nearly identical degradation.”
“Ideally, however, the UDR obtained from different patches or the entire image should exhibit consistency.”
- However, degradation in real-world images is often not similar in different local regions. This is because image degradation is affected by a variety of factors, and the way these factors are distributed and act in an image may vary from region to region.
For example.
non-uniform motion blur produces different blurring effects in different regions.
Noise is not uniform in distribution and intensity in an image, similarly, noise characteristics may vary from region to region.
Compression may vary from region to region depending on the compression rate and image content.

(3) Non-reference metrics can be considered to add for convincing enhancement, similar to SeeSR (CVPR2024), given that some of the methods in Table 1 are GAN-based, it is not fair to compare PSNR/SSIM.

(4) In Table 1, for the datasets div2k, bsds100, urban100, t91, they are first synthesizing low-resolution images via degradation process of RealESRGAN and then used for testing? dped and realSR are tested directly using the low-resolution images?

(5) The references are confusing and for the conference reference, four different versions appear:
- In ICCV.
- In The IEEE Conference on Computer Vision and Pattern Recognition (CVPR).
- In Proceedings of the IEEE/CVF Conference on Computer Vision and Pattern Recognition.
- In Computer Vision–ECCV 2020: 16th European Conference, Glasgow, UK, August 23–28, 2020, Proceedings, Part XXII 16. Springer, 272–289.

(6) Some arXiv references can be replaced with published versions.

**Suitability:**

2

---

### Official Review · Reviewer_8Stg · 2024-05-24

**Rating:** 2
**Confidence:** 4

**Summary:**

The paper proposes an \textbf{U}ncertainty-based degradation representation for blind \textbf{S}uper-\textbf{R}esolution framework (\textbf{USR}). By suppressing the uncertainty of local degradation representations in images, USR facilitated self-supervised learning of degradation representations. The USR consists of two components: Adaptive Uncertainty-Aware Degradation Extraction (AUDE) and a feature extraction network composed of Variable Depth Dynamic Convolution (VDDC) blocks.

**Strengths:**

The topic is of interesting.

**Limitations:**

1)	In equation (3), for the joint probability is seen as independent of each other, but for x1 and x2, taken from different regions of the same image with the same degradation pattern, there will be mutual influence. Why it can be seen to be independent of each other?
2)	The USR method performs well on other datasets but is more than 2 db lower than DCLS on T91, which is also a natural image and should perform similarly to datasets such as Urban100. What is the main reason ? Please explain it.
3)	In equation (5), the mean and variance are output from the network parametered by W. How should this be defined, and how does one go about outputting the mean and variance?
4)	How does Uncertainty Aware weight reduce the uncertainty of the degenerate representation in Eq. (9), and why is it a subtraction of the canonical term for Eq. (11)?
5)	Is the dataset generated using Real-ESRGAN during the training of the network, and how about the number of parameters for different algorithms?
6)	In the ablation experiment in Table2, several cases are missing. The effect of the model not used by either AUDE or AIS should be given. What is the case in Table 3, where the number of VDDC is equal to 9?

**Suitability:**

2

---

### Official Review · Reviewer_36U5 · 2024-05-24

**Rating:** 4
**Confidence:** 2

**Summary:**

This work introduces an uncertainty-based degradation representation for blind super-resolution framework. To quantify and mitigate the uncertainty in uncertainty-based degradation representation estimation, the work proposes the self-supervised uncertainty contrast to suppress the uncertainty of local degradation representations in images. Extensive experiments demonstrate the effectiveness of the proposed method.

**Strengths:**

- It proposes variable depth dynamic convolution blocks and adaptive uncertainty-aware degradation extraction in the SR process.
- Multiple datasets have been validated.
- The PSNR/SSIM is promising.

**Limitations:**

- The motivation of the work is not presented clearly. Why we should suppress the uncertainty and how to achieve it are not shaped well.
- The results in Fig.3 show limited improvement of the proposed method. Even, from my view, the proposed method is not good as for the first image in the figure.

**Suitability:**

2

---

### Meta-Review · Area_Chair_v29E · 2024-07-04

**Recommendation:** Accept (Poster)
**Confidence:** 5

**Metareview:**

The paper received mixed reviews (2x Borderline Accept, 1x Weak Reject) and the authors provided a response to the raised concerns.

Reviewer 8Stg (Weak Reject) appreciates that the authors addressed most of his concerns, however, still has concerns with respect to reported results (Table 1, Fig.3) and demands reproducibility codes and explanation.

The other reviewers recommend acceptance appreciating the contributions and that "the performance is charming".

After carefully reading the paper, the reviews, the authors' response and the reviewers' final justifications, the Meta-Reviewer agrees that the paper indeed makes significant contributions to a challenging problem -- blind super-resolution, and achieves promising results. At the same time, in Fig.3 the visual result is indeed questionable and the authors should make sense out of it and the provided authors' response contains important information that should be in the paper.

The paper is worthy for publication and the authors are invited to refine their work based on the received feedback and integrate relevant contents from their response to the camera ready paper.

Also, the authors are encouraged to release codes for reproducibility.